# DENSE RECURRENT NEURAL NETWORK
# WITH ATTENTION GATE

## ABSTRACT

We propose the dense RNN, which has fully connections from each hidden state directly to multiple preceding hidden states of all layers. As the density of the connection increases, the number of paths through which the gradient flows is increased. The magnitude of gradients is also increased, which helps to prevent the vanishing gradient problem in time. Larger gradients, however, may cause exploding gradient problem. To complement the trade-off between two problems, we propose an attention gate, which controls the amounts of gradient flows. We describe the relation between the attention gate and the gradient flows by approximation. The experiment on the language modeling using Penn Treebank corpus shows dense connections with the attention gate improve the model's performance of the RNN.

## 1 INTRODUCTION

In order to analyze sequential data, it is important to choose an appropriate model to represent the data. Recurrent neural network (RNN), as one of the model capturing sequential data, has been applied to many problems such as natural language (Mikolov et al., 2013), machine translation (Bahdanau et al., 2014), speech recognition (Graves et al., 2013). There are two main research issues to improve the RNNs performance: 1) vanishing and exploding gradient problems and 2) regularization.

The vanishing and exploding gradient problems occur as the sequential data has long-term dependency (Hochreiter, 1998; Pascanu et al., 2013). One of the solutions is to add gate functions such as the long short-term memory (LSTM) and gated recurrent unit (GRU). The LSTM has additional gate functions and memory cells (Hochreiter & Schmidhuber, 1997). The gate function can prevent the gradient from being vanished during back propagation through time. Gated recurrent unit (GRU) has similar performance with less gate functions (Chung et al., 2014).

The part of sequential data whose boundary to distinguish the consecutive other parts, has the hierarchical structures. To handle the hierarchy, the model should capture the multiple timescales. In hierarchical multiple recurrent neural network (HM-RNN, Chung et al. (2016)), the boundary information is also learned by implementing three operations such as update, copy and flush operator. In clockwork RNN (Koutnik et al., 2014), the hidden states are divided into multiple sub-modules, which act with different periods to capture multiple timescales. As all previous states within the recurrent depth do not always affect the next state, memory-augmented neural network (MANN, Gulcehre et al. (2017)) uses the memory to remember previous states and retrieve some of previous states if necessary.

The basic way to handle multiple timescales along with preventing the vanishing gradient problem is to increases both of feedforward depth and recurrent depth to capture multiple timescales. Feedforward depth is the longest path from the input layer to the output layer. Recurrent depth is the longest path from arbitrary hidden state at time $t$ to same hidden sate at time $t + t'$ (Zhang et al., 2016). Increasing feedforward depth means stacking multiple recurrent layers deeply. It can capture fast and slow changing components in the sequential data (Schmidhuber, 1992; El Hihi & Bengio, 1996; Hermans & Schrauwen, 2013). The low level layer in the stacked RNN captures short-term dependency. As the layer is higher, the aggregated information from lower layer is abstracted. Thus, as the layer is higher, the capacity to model long-term dependency increases. The number of nonlinearities

in the stacked RNN, however, is proportional to the number of unfolded time steps regardless of the feedforward depth. Thus, the simple RNN and stacked RNN act identically in terms of long run.

Increasing recurrent depth also increases the capability to capture long-term dependency in the data. The hidden state in vanilla RNN has only connection to previous time step's hidden state in the same layer. Adding the connections to multiple previous time steps hidden states can make the shortcut paths, which alleviates the vanishing problem. Nonlinear autoregressive exogenous model (NARX) handles the vanishing gradient problem by adding direct connections from the distant past in the same layer (Lin et al., 1996). Similarly, higher-order RNN (HO-RNN) has the direct connections to multiple previous states with gating to each time step (Soltani & Jiang, 2016). Unlike other recurrent models that use one connection between two consecutive time steps, the recurrent highway network (RHN) adds multiple connections with sharing parameters between transitions in the same layer (Zilly et al., 2016).

The vanilla RNN has only one path connected with previous hidden states. Thus, it is hard to apply standard dropout technique for regularization as the information is being diluted during training of long-term sequences. By selecting the same dropout mask for feedforward, recurrent connections, respectively, the dropout can apply to the RNN, which is called a variational dropout (Gal & Ghahramani, 2016).

This paper proposes a dense RNN that has both of feedforward and recurrent depths. The stacked RNN increases the complexity by increasing feedforward depth. NARX-RNN and HO-RNN increase the complexity by increasing recurrent depth. The model with the feedforward depth can be combined with the model with the recurrent depth, as the feedforward depth and recurrent depth have an orthogonal relationship. Gated feedback RNN has the fully connection between two consecutive timesteps. As the connection of gated feedback is not overlapped with the model with orthogonal depths, all three features, adding feedforward depth, recurrent depth, and gated feedback, can be modeled jointly . With the three features, we propose the attention gate, which controls the flows from each state so that it enhances the overall performance.

The contributions of this paper are summarized: 1) dense RNN that is aggregated model with feedforward depth, recurrent depth and gated feedback function, 2) extension of the variational dropout to the dense RNN.

## 2 BACKGROUND

There are largely two methods to improve the performance of RNN. One is to extend previous model by stacking multiple layers or adding gate functions. The other is using regularization such as dropout to avoid overfitting.

### 2.1 EXTENSION OF RECURRENT NEURAL NETWORK MODELS

In simple recurrent layer, $h_t$, the hidden state at time t, is a function of input $x_t$ and preceding hidden state $h_{t-1}$, which is defined as follows:

$$h_t = \phi(h_{t-1}, x_t) = \phi(W x_t + U h_{t-1}) \tag{1}$$

where $U$ and $W$ are respectively the feedforward and recurrent weight matrix and $\phi$ means an element-wise nonlinear function such as $Tanh$. In simple recurrent layer, the last hidden state $h_{t-1}$ has to memorize all historical inputs. As the memorizing capacity of the hidden state is limited, it is hard to capture long-term dependency in sequential data. Stacked recurrent neural network, stacked of the simple recurrent layers can capture the long-dependency, which is defined as follows:

$$h_t^j = \phi(W^j h_t^{j-1} + U^{j \to j} h_{t-1}^j) \tag{2}$$

where $W^j$ is the weight matrix for transition from layer $j - 1$ to $j$ and $U^j$ is the weight matrix for transition from in timestep $t - 1$ to timestep $t$ at layer $j$. The stacked RNN can model multiple timescales of the sequential data. As the information travels toward upper layer, the hidden state can memorize abstracted information of sequential data that covers more long-term dependency.

The other way to capture the long term dependency is to increase the recurrent depth by connecting the hidden state at timestep $t$ directly to multiple preceding hidden states (Soltani & Jiang, 2016),

which is defined as follows:

$$h_t^j = \phi(W^j h_t^{j-1} + \sum_{k=1}^{K} U^{(k,j)\to j} h_{t-k}^j) \tag{3}$$

where $U^{(k,j)\to j}$ is the weight matrix from layer $j$ at timestep $t-k$ to layer $j$ at timestep $t$, and $K$ is the recurrent depth. The direct connections make the shortcut paths from preceding multiple hidden states. Compared with the model without shortcut paths, the model with shortcut paths enables to access preceding hidden states further way from $h_t^j$ with same number of transitions.

Most recurrent models have only recurrent connections between hidden states with same layers. By adding feedback connections to the hidden state $h_t^j$ from the preceding hidden states $h_{t-1}^i$ at different layer of $h_t^j$, the model adaptively captures the multiple timescales from long-term dependency, which is defined as follows:

$$h_t^j = \phi(W^j h_t^{j-1} + \sum_{i=1}^{L} U^{i\to j} h_{t-1}^i) \tag{4}$$

where $U^{i\to j}$ is the weight matrix from layer $i$ at timestep $t-1$ to layer $j$ at timestep $t$, and $L$ is the feedforward depth. To control the amount of flows between different hidden states with different time scales (Chung et al., 2014), the global gate is applied as follows:

$$h_t^j = \phi(W^j h_t^{j-1} + \sum_{i=1}^{L} g^{i\to j} U^{i\to j} h_{t-1}^i). \tag{5}$$

In (5), $g^{i\to j}$ is the gate to control the information flows from each hidden state to multiple preceding hidden states of all layers, which is defined as follows:

$$g^{i\to j} = \sigma(w_g h_t^{j-1} + u_g^{i\to j} h_{t-1}^*) \tag{6}$$

where $w_g^j$ is a vector whose dimension is same with $h_t^{j-1}$, $u_g^{i\to j}$ is a vector whose dimension is same with $h_{t-1}^*$ that is a concatenated vector of all hidden states from previous time step, and $\sigma$ is an element-wise sigmoid function.

Gated feedback function is also applied to LSTM. In gated LSTM, input gate $i_t^j$, forget gate $f_t^j$, output gate $o_t^j$, and memory cell gate $\tilde{c}_t^j$ are defined as follows:

$$i_t^j = \sigma(W_i^j h_t^{j-1} + U_i^{j\to j} h_{t-1}^j), \tag{7a}$$

$$f_t^j = \sigma(W_f^j h_t^{j-1} + U_f^{j\to j} h_{t-1}^j), \tag{7b}$$

$$o_t^j = \sigma(W_o^j h_t^{j-1} + U_o^{j\to j} h_{t-1}^j), \tag{7c}$$

$$\tilde{c}_t^j = \phi((W_c^j h_t^{j-1} + \sum_{i=1}^{L} g^{i\to j} U_c^{i\to j} h_{t-1}^j). \tag{7d}$$

Compared with conventional LSTM, the gated feedback LSTM has gated feedback function in the memory cell gate $\tilde{c}_t^j$. Through gates and memory cell state in (7a)-(7d), the new memory cell state $c_t^j$ and hidden state $h_t^j$ respectively are calculated as follows:

$$c_t^j = f_t^j \cdot c_{t-1}^j + i_t^j \cdot \tilde{c}_t^j, \tag{8a}$$

$$h_t^j = o_t^j \cdot \phi(c_t^j) \tag{8b}$$

where the dot product means element-wise multiplication.

## 2.2 Dropout of Recurrent Neural Network

As dropout is one method of neural network regularization, it prevents the model from being over-fitted to training set. However, it is hard to apply the standard dropout to recurrent connections. As the sequence is longer, the information is affected by the dropout many time during backpropagation through time, which makes memorizing long sequences hard. Thus, applying the standard dropout only to feedforward connections is recommended (Zaremba et al., 2014). It is expressed as follows:

$$h_t^j = \phi(W^j D_t^j(h_t^{j-1}) + U^{j \to j} h_{t-1}^j) = \phi(W^j(m_t^{j-1} \cdot h_t^{j-1}) + U^{j \to j} h_{t-1}^j) \tag{9}$$

where $D_t^j$, as a dropout operator for every time step, makes $h_t^{j-1}$ being masked with Bernoulli dropout mask $m_t^{j-1}$ randomly generated for every time step. Moon et al. (2015) proposed how to apply the dropout to recurrent connections efficiently. By considering the whole sequential data as one input at a sequential level, same dropout mask is applied to recurrent connections at all time steps during training, which is expressed as follows:

$$h_t^j = \phi(W^j D_t^j(h_t^{j-1}) + U^{j \to j} D^{j \to j}(h_{t-1}^j)) = \phi(W^j(m_t^{j-1} \cdot h_t^{j-1}) + U^{j \to j}(m^j \cdot h_{t-1}^j)) \tag{10}$$

where $D^{j \to j}$, as a time-independent dropout operator, makes $h_{t-1}^j$ being masked with Bernoulli dropout mask $m^j$ randomly generated regardless of time.

Gal & Ghahramani (2016) applied variational dropout to RNN and proved the relation between Bayesian inference and dropout theoretically. Variational dropout in RNN applies same masks regardless of time to feedforward connections, similar to recurrent connections, which is expressed as follows:

$$h_t^j = \phi(W^j D^j(h_t^{j-1}) + U^{j \to j} D^{j \to j}(h_{t-1}^j)) = \phi(W^j(m^{j-1} \cdot h_t^{j-1}) + U^{j \to j}(m^j \cdot h_{t-1}^j)). \tag{11}$$

# 3 Dense Recurrent Neural Network

The skip connections that bypass some layers enables deep networks to be trained better than the models without skip connections. Many research (He et al., 2016; Huang et al., 2016) uses skip connections toward feedback connections. In this paper, we apply the skip connections toward recurrent connections. The shortcut paths from preceding multiple hidden states to the hidden state at time $t$ is equal to the skip connections through time. The shortcut paths include the feedback connections between different layers of different timesteps. Each connection is controlled by the attention gate, similar to the global gate in the gated feedback RNN. The dense RNN is defined as follows:

$$h_t^j = \phi(W^j h_t^{j-1} + \sum_{k=1}^{K} \sum_{i=1}^{L} g^{(k,i) \to j} U^{(k,i) \to j} h_{t-k}^i) \tag{12}$$

where $g^{(k,i) \to j}$ is the attention gate, which is defined as follows:

$$g^{(k,i) \to j} = \sigma(w_g^i h_t^{j-1} + u_g^{(k,i) \to j} h_{t-k}^i). \tag{13}$$

(13) is a function of preceding hidden state at layer $i$ and time $t - k$, while (6) is a function of concatenated all preceding hidden states. The intuition why we change the gate function and we call it to attention gate is described in Section 3.1.

The dense RNN can be easily extended to dense LSTM. The dense LSTM is defined as follows:

$$i_t^j = \sigma(W_i^j h_t^{j-1} + \sum_{k=1}^{K} \sum_{i=1}^{L} g_i^{(k,i) \to j} U_i^{(k,i) \to j} h_{t-k}^i), \tag{14a}$$

$$f_t^j = \sigma(W_f^j h_t^{j-1} + \sum_{k=1}^{K} \sum_{i=1}^{L} g_f^{(k,i) \to j} U_f^{(k,i) \to j} h_{t-k}^i), \tag{14b}$$

$$o_t^j = \sigma(W_o^j h_t^{j-1} + \sum_{k=1}^{K} \sum_{i=1}^{L} g_o^{(k,i) \to j} U_o^{j \to j} h_{t-1}^j), \tag{14c}$$

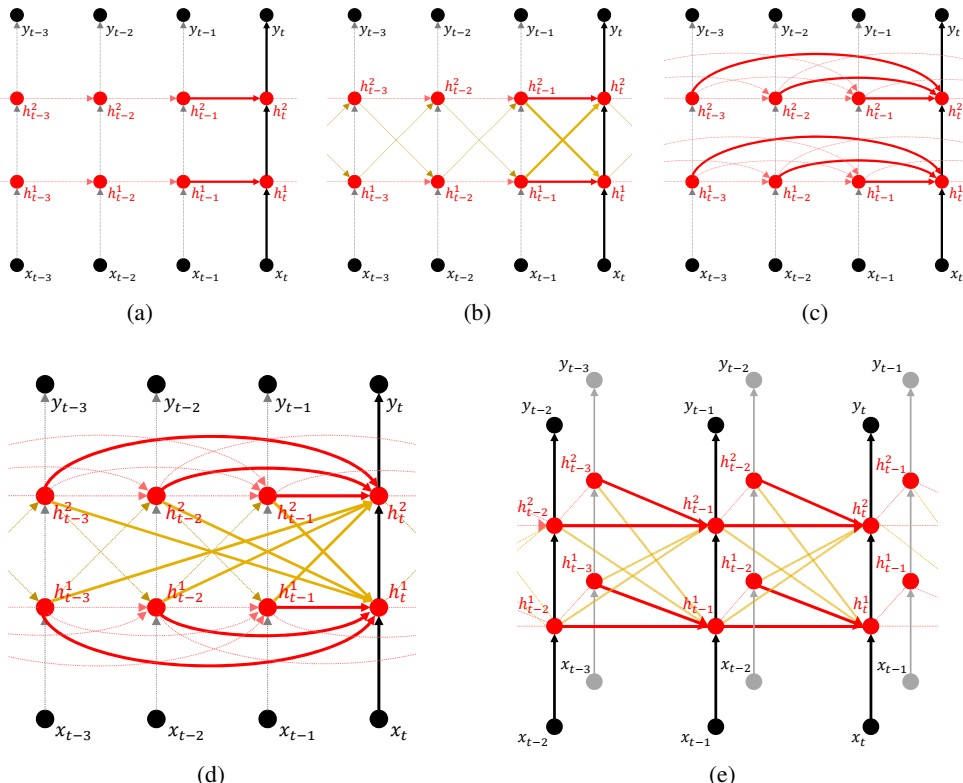

Figure 1: (a) Conventional RNN unfolded in time. (b) Gated feedback RNN. (c) RNN with connections across multiple preceding states. (d) Dense RNN integrated with (b) and (c). Hidden states are represented in red. The connections used in current steps feedforward are highlighted in bold. The feedback connections between upper and lower layers are represented in yellow. (e) Unfolded Dense RNN in both of time and the number of preceding states.

$$\tilde{c}_t^j = \phi((W_c^j h_t^{j-1} + \sum_{k=1}^{K} \sum_{i=1}^{L} g_c^{(k,i) \to j} U_c^{i \to j} h_{t-1}^j). \tag{14d}$$

Unlike gated feedback LSTM, the attention gate $g$ is applied to all gates and memory cell state.

We analyze the advantages of dense connections intuitively and theoretically. In addition, we propose the dropout method for dense connections.

## 3.1 INTUITIVE ANALYSIS

Recurrent connections enable to predict next data given previous sequential data. In the language modeling, the RNNs can predict next word based on the last word and the last context accumulated before the last word. It assumes only last word affect to predict next word. For instance, "the sky is" is given from the full sentence "the sky is blue" and the goal is to predict the word "blue". In this case, the preceding word "sky provides the better clue than the preceding word "is". Inspired by the fact, we propose the dense model that predicts the next word by directly referring to recent preceding words. In other words, the output $h_t^j$ is a function of input $h_t^{j-1}$ and recent preceding output $h_{t-k}^j$ as in (3).

The higher the layer in a neural network, the more abstract the hidden states. In language modeling, hidden states represent the characteristics of words, sentences, and paragraphs as the layer increases. The conventional RNN has only the connection between same layer. It means the preceding words, sentences, paragraphs determine next words, sentences and paragraphs, respectively. The given

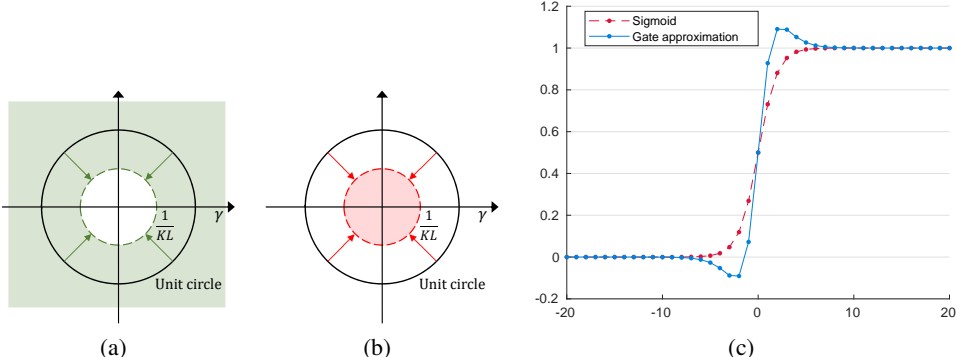

(a)          (b)          (c)

Figure 2: (a) The stable region from exploding gradient problem. The attention gate increases the stable region from 1 to $1/(KL)$. (b) The stable region from vanishing gradient problem. The attention gate decreases the stable region from 1 to $1/(KL)$. (c) The approximation of the attention gate to determine vanishing and exploding gradient boundary.

word, however, can also determine the context of next paragraph. Also, the given paragraph can determine next word. For instance, the word "mystery" in "it is mystery" can be followed by the paragraph related to "mystery" and vice versa. The feedback connections can reflect the fact.

In (4), preceding words, sentences, and paragraphs affects next words, sentences, and paragraphs with same scale. Preceding words, however, dont affects next word prediction evenly. In the sentence "The sky is blue", the word "sky" has a very close relation with the word "blue". The word "The", as an article, has a less relation with the word "blue". The amount how two words are related depends on the kind of the two words. We define the the degree of relevance as gated attention $g$ as in (5).

The attention $g$ is determined by the preceding word itself and the last word given as input. In the sentence "The sky is blue", the features of the word "the", and "sky" denote $h_{t-2}$, and $h_{t-1}$, respectively and the word "is" denotes $x_t$ or $h_t^0$. Then, the attention to predict the word"blue" from the word "The" is determined by the word "The and "blue". The attention to predict the word "blue" from the word "is" is determined by the word "is" and "blue". In other words, the attention is dependent on the previous hidden state $h_{t-k}^j$ and input $h_t^{j-1}$ at certain time step as in (13).

## 3.2 THEORETICAL ANALYSIS

The vanishing and exploding gradient problems happen the sequential data has long term dependency. During backpropagation through time, error $E_T$'s gradient with respect to the recurrent weight matrix $U^j$ is vanished or exploded as the sequence is longer, which is expressed as follows:

$$\frac{\partial E_T}{\partial U^j} = \sum_{t=1}^{T} \frac{\partial E_T}{\partial U^j} = \sum_{t=1}^{T} \frac{\partial E_T}{\partial h_T^j} \frac{\partial h_T^j}{\partial h_t^j} \frac{\partial h_t^j}{\partial U^j} = \sum_{t=1}^{T} \frac{\partial E_T}{\partial h_T^j} \left( \prod_{\tau=t+1}^{T} \frac{\partial h_\tau^j}{\partial h_{\tau-1}^j} \right) \frac{\partial h_t^j}{\partial U^j}. \tag{15}$$

The critical term related to vanishing and exploding gradient problems is $\partial h_\tau^j / \partial h_{\tau-1}^j$. To find the relation between vanishing and exploding gradient problems and dense connections, we assume the worst situation in which the vanishing and exploding gradient problem may arise. The term $\partial h_\tau^j / \partial h_{\tau-1}^j$ denotes $A^j$. If the $A_{max}^j$ is less than 1, the gradient with respect to $U^j$ would be exploded and if the $A_{min}^j$ is greater than 1, gradient with respect to $U^j$ would be vanished. In dense recurrent network, there are more paths to flow the gradients. The $A^j$ in dense recurrent network is approximated as follows:

$$A^j = \sum_{k=1}^{K} \sum_{i=1}^{L} \frac{\partial h_{\tau+k-1}^i}{\partial h_{\tau-1}^j} = \sum_{k=1}^{K} \sum_{i=1}^{L} A^{(k,i) \to j} \tag{16}$$

where the superscript $(k, i) \to j$ means the direction of the path from $h^i_{\tau+k-1}$ to $h^j_\tau$. Assuming $A^{(k,i)\to j}$ is $A^{(k,i)\to j}_{max}$, $A^j_{max}$ is $KLA^{(k,i)\to j}_{max}$, which reduces the vanishing gradient boundary from 1 to $1/(KL)$ as shown in Figure 2(a).

Though dense connections are able to alleviate the vanishing gradient problem, it causes the gradient exploding. To alleviate the problem caused by dense connection, we add the attention gates as in (13). The attention gate $g^{(k,i)\to j}$ can control the magnitude of $A^{(k,i)\to j}$, which is expressed as follows:

$$A^j = \sum_{k=1}^{K}\sum_{i=1}^{L} \frac{\partial h^i_{\tau+k-1}}{\partial h^j_{\tau-1}} \approx \sum_{k=1}^{K}\sum_{i=1}^{L} g^{(k,i)\to j} \cdot A^{(k,i)\to j}. \tag{17}$$

In (17), the $g^{(k,i)\to j}$ is trainable so that the vanishing and exploding boundary is determined adaptively. In dense RNN with attention gates, $h^i_{\tau+k-1}$ is expressed as follows:

$$h^i_{\tau+k-1} = \phi(g^{(k,i)\to j} \cdot U^{(k,i)\to j}h^j_{\tau-1} + \theta) \tag{18}$$

where $\theta$ is not relevant parameters with $h^j_{\tau-1}$, $g^{(k,i)\to j}$ is $\sigma(U^{(k,i)\to j}_g h^j_{\tau-1})$. For simplicity, (18) is expressed as $y = \phi(g(U_g x) \cdot U x + \theta)$. Gradient of $y$ with respect to $x$ is calculated as follows:

$$\begin{aligned}
\frac{\partial y}{\partial x} &= y(1-y)[g(1-g)U_g U x + gU] \\
&= y(1-y)U[g(1-g)U_g x + g] = \tilde{g}y(1-y)U.
\end{aligned} \tag{19}$$

The (19) is scaled with $\tilde{g}$ compared to $\frac{\partial y}{\partial x} = y(1-y)U$ without attention gate $g$. As $g$ and $\tilde{g}$ are similar as shown in Figure 2(c), $\tilde{g}$ is approximated $g$ as in (19).

In recurrent highway network (RHN, Zilly et al. (2016)), the effect of highway was described using the Geršgorin circle theorem (GST, Geršhgorin (1931)). Likewise, the dffect of the attention gate in the proposed model can be interpreted using GST. For simplicity, we only formulate recurrent connection with omitting feedforward connection, $h_{t+1} = \phi(U h_t)$. Then, the Jacobian matrix $A = \frac{\partial h_{t+1}}{\partial h_t}$ is $U^T \text{diag}[\phi'(U h_t)]$. By letting $\gamma$ be a maximal bound on $\text{diag}[\phi'(U h_t)]$ and $\rho_{max}$ be the largest singular value of $U^T$, the norm of the Jacobian satisfies using the triangle inequality as follows:

$$\|A\| \le \|U^T\| \left\| \text{diag}[\phi'(U h_t)] \right\| \le \gamma\rho_{max}. \tag{20}$$

The value $\gamma\rho_{max}$ is less than 1, the vanishing gradient problem happens and the $\|A\|$ is greater than 1, the exploding gradient problem happens as the range of the $\|A\|$ has no explicit boundary. The spectrum of $A$, the set of $\lambda$ in $A$, is evaluated as follows:

$$\text{spec}(A) \subset \bigcup_{i\in 1,\ldots,n} \{\lambda \in \mathbb{C} | \|\lambda - a_{ii}\|_{\mathbb{C}} \le \sum_{j, j\neq i}^{n} |a_{ij}|\}, \tag{21}$$

which means the eigenvalue $\lambda$ lies on the circle whose radius is the summation of abstract values of all elements except the diagonal element $a_{ii}$ and center is the diagonal element $a_{ii}$. The simplified recurrent connection with the attention gate is $h_{t+1} = \phi(g U h_t)$ where $g$ is $U_g, h_t$. Then, the Jacobian matrix $A = \frac{\partial h_{t+1}}{\partial h_t}$ is expressed as follows:

$$A = \left(g + g'\text{diag}(U_g)\text{diag}(h_{t,i})\right)U^T\text{diag}[\phi'(g U h_t)]. \tag{22}$$

The spectrum of (22) is expressed as follows:

$$\text{spec}(A) \subset \bigcup_{i\in 1,\ldots,n} \{\lambda \in \mathbb{C} | \|\lambda - (g + g'U_{g,i}h_{t,i})a_{ii}\|_{\mathbb{C}} \le \sum_{j, j\neq i}^{n} |ga_{ij}|\}. \tag{23}$$

The scaled term term $g + g'U_{g,i}h_{t,i}$ in (23) can be approximated as $g$ as shown in Figure 2(b). Thus, the upper bound of $\|A\|$ is approximately less than 1 so that the attention gate $g$ can alleviate the exploding problem.

Table 1: Perplexity on the Penn Treebank language modeling task.

| Network | # Hidden size | # Feed forward | # Recurrent | # Param. | Valid | Test |
|---|---|---|---|---|---|---|
| LSTM (Zaremba et al., 2014) | 200 | 2 | 1 | 5M | 104 | 100.4 |
| | 650 | 2 | 1 | 20M | 86.2 | 82.7 |
| | 1500 | 2 | 1 | 66M | 82.2 | 78.4 |
| VD-LSTM+RE (Inan et al., 2016) | 200 | 2 | 1 | 2.65M | 89.9 | 85.1 |
| | 650 | 2 | 1 | 8.6M | 77.4 | 74.7 |
| | 1500 | 2 | 1 | 19.8M | 74.5 | 71.2 |
| Dense LSTM (2 hidden layers) | 200 | 2 | 1 | 3.0M | 85.39 | 80.35 |
| | 200 | 2 | 2 | 3.6M | 83.39 | 78.64 |
| | 200 | 2 | 3 | 4.3M | 83.56 | 79.05 |
| | 200 | 2 | 4 | 4.9M | 83.28 | 78.82 |
| | 200 | 2 | 5 | 5.6M | 83.70 | 78.99 |
| Dense LSTM (3 hidden layers) | 200 | 3 | 1 | 3.9M | 83.12 | 78.95 |
| | 200 | 3 | 2 | 5.4M | 83.08 | 78.67 |
| | 200 | 3 | 3 | 6.9M | 82.38 | 78.15 |
| | 200 | 3 | 4 | 8.3M | 82.56 | 77.91 |
| | 200 | 3 | 5 | 9.8M | 82.98 | 78.69 |

### 3.3 EXTENSION OF VARIATIONAL DROPOUT

In dense RNN, as recurrent depth increases, the number of parameters also increases, which makes the model vulnerable to overfitting to training dataset. To prevent the model from overfitting to training dataset, the dropout is applied. The variational dropout, proved to show good performance in the RNN models, uses same random masks at every time step for feedforward connections, and recurrent connections, respectively. In implementation of variational dropout, each state is dropped with the random mask, which is followed by weighted sum. This concept is extensively applied to dense RNN so that the same random mask is used regardless of time and recurrent depth. Extension of variational dropout to the dense connection is expressed as follows:

$$h_t^j = \phi(W^j(m^{j-1} \cdot h_t^{j-1}) + \sum_{k=1}^{K} \sum_{i=1}^{L} g^{(k,i) \to j} U^{(k,i) \to j}(m^i \cdot h_{t-k}^i)). \tag{24}$$

## 4 EXPERIMENT

In our experiment, we used Penn Tree corpus (PTB) for language modeling. PTB consists of 923k training set, 73k validation set, and 82k test set. As a baseline model, we select the model proposed by Zaremba et al. (2014), which proposed how to predict next word based on the previous words. To improve the accuracy, Zaremba et al. (2014) proposed regularization method for RNN.

The baseline models hidden sizes are fixed as 200, 650, and 1500, which are called as small, medium, and large network. In this paper, we fixed hidden size as 200 for all hidden layers with varying the feedforward depth from 2 to 3, recurrent depth from 1 to 5. The word embedding weights were tied with word prediction weights.

To speed up the proposed method, all of matrix multiplications in (12) was performed by the batch matrix-matrix product of two matrices. Each of batches is rewritten as follows:

$$\tilde{h}_{t-k}^{j \to i} = U^{(k,i) \to j} h_{t-k}^i. \tag{25}$$

The (12) is rewritten as follows:

$$h_t^j = \phi(W^j h_t^{j-1} + \sum_{k=1}^{K} \sum_{i=1}^{L} g^{(k,i) \to j} \tilde{h}_{t-k}^i). \tag{26}$$

The proposed dense model is trained with stochastic gradient decent. The initial rate was set to 20, which was decayed with 1.1 after the epoch was over 12. The training was terminated when the epoch reaches 120. To prevent the exploding gradient, we clipped the max value of gradient as 5.

As a regularization, we adopt variational dropout, which uses the random masks regardless of time. We configured the word embeddings dropout to 0.3, feedforward connections dropout 0.2, and recurrent connections dropout rate varies from 0.2 to 0.5. The Table  1, as a trained result, compares the baseline model, RNN model with variational dropout and using same word space. In dense RNN, the perplexity is better than two models. The best models recurrent depth is 2 and the perplexity of valid set is 83.28 and that of test set is 78.82.

## 5    CONCLUSION

This paper proposed dense RNN, which has fully connections from each hidden state to multiple preceding hidden states of all layers directly. Each previous hidden state has its attention gate that controls the amount of information flows. To evaluate the effect of dense connections, we used Penn Treebank corpus (PTB). The result of dense connection was confirmed by varying the recurrent depth with the attention gate. The dense connections with the attention gate made the model's perplexity less than conventional RNN.

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
