# OpenReview forum: "Dense Recurrent Neural Network with Attention Gate"
_ICLR.cc/2018/Conference — Reject_

### Official Review · AnonReviewer2 · 2017-11-27

**Rating:** 2
**Confidence:** 4

**Review:**

 This paper proposes a new type of RNN architectures called Dense RNNs. The authors combine several different RNN architectures and claim that their RNN can model long-term dependencies better, can learn multiscale representation of the sequential data, and can sidestep the exploding or vanishing gradients problem by using parametrized gating units.

Unfortunately, this paper is hard to read, it is difficult to understand the intention of the authors. The authors make several claims without any supportive reference or experimental evidence. Both intuitive and theoretical justifications of the proposed architecture are not so convincing. The experiment is only done on PTB dataset, and the reported numbers are not that promising either.

This paper tries to combine three different features from previous works, and unfortunately, it is not so well conducted.

---

### Official Review · AnonReviewer1 · 2017-11-27
**Dense RNNs with Attention Gate**

**Rating:** 4
**Confidence:** 4

**Review:**

Summary:

This paper proposes a fully connected dense RNN architecture that has connections to every layer and the preceding connections of each layer. The connections are also gated by using a simple gating mechanism. The authors very briefly discusses about the effect of these on the dynamics of the learning. They report results on PTB character-level language modelling task.


Questions:
What is the computational complexity of this approach compared to a vanilla RNN architecture?
What is the implications of these skip connections in terms of memory consumption during BPTT?
Did you use gradient clipping and have you used any specific type of initialization for the parameters?
How would this approach would compare against the Clockwork RNNs which has a block-diagonal weight matrices? [1]
How would dense-RNNs compare against to the MANNs [2]?
How would you implement this model efficiently?

Pros:
Interesting idea.
Cons:
Lack of experiments and empirical results supporting the arguments.
Hand-wavy theory.
Lack of references to the relevant literature.

General Comments:
In general the paper is relatively well written despite having some minor typos. The idea is interesting, however the experiments in this paper is seriously lacking. The only results presented in this paper is on PTB. The results are quite behind the SOTA and PTB is a really tiny, toyish language modeling task. The theory is very hand-wavy, the connections to the previous attempts to come up with related properties of the recurrent models should be cited. The Figure 2 is very related to the Gersgorin circle theorem in [3]. The discussion about the skip-connections is very related to the results in [2].

Overall, I think this paper is rushed and not ready for the publication.

[1] Koutnik, J., Greff, K., Gomez, F., & Schmidhuber, J. (2014, January). A clockwork rnn. In International Conference on Machine Learning (pp. 1863-1871).
[2] Gulcehre, Caglar, Sarath Chandar, and Yoshua Bengio. "Memory Augmented Neural Networks with Wormhole Connections." arXiv preprint arXiv:1701.08718 (2017).
[3] Zilly, Julian Georg, Rupesh Kumar Srivastava, Jan Koutník, and Jürgen Schmidhuber. "Recurrent highway networks." arXiv preprint arXiv:1607.03474 (2016).

---

> ### Author Response · Authors · 2018-01-05
> **Response to AnonReviewer1**
>
>  We thanks the reviewers for their work. And your review was very helpful for me.
> I answered your questions and based on the aswers, I updated my paper.
>
> Q. What is the computational complexity of this approach compared to a vanilla RNN architecture?
>
> A. The number of parameters in dense rnn is feedforward depth^2 * recurrent depth * hidden size^2
> The number of parameters in vanilla rnn is hidden size^2.
>
> Doubling the hidden size and doubling feedforward depth have same effect in terms of the number of parameters. And doubling recurrent depth is more efficient than doubling of hidden size with same factor.
>
> Q. What is the implications of these skip connections in terms of memory consumption during BPTT?
>
> A. If there is no skip connections, the gradients have to flow with stopping by every hidden states, it makes the parameters being vanished or exploded. The skip connections make the gradients pass the less number of hidden states, it alleviates the vanishing gradient or exploding gradient problems.
>
> Q. Did you use gradient clipping and have you used any specific type of initialization for the parameters?
>
> A. We used gradient clipping with the value 5. We used stochastic gradient optimizer with sceduling the learning rate.
>
> Q. How would this approach compare against the Clockwork RNNs which has a block-diagonal weight matrices?
>
> A. In clockwork RNN, the hidden states are divided into multiple sub-modules, which act with different periods to capture multiple timescales. In dense RNN, all previous states within recurrent depth affect current hidden state every time step. The periods underlying the sequences are automatically selected using the attention gate in dense RNN. In summary, clockwork RNN pre-defines the frequency to capture from the sequence and dense RNN learns the frequency using the attention gate.
>
> Q. [1] How would dense-RNNs compare against to the MANNs [2]?
>
> A. All previous states within the recurrent depth don't always affect the next state. Thus, MANN uses the memory to remember previous states and retrieve some of previous states if necessary. This is similar concept. However, the MANN has only connections between same layers.
>
> Q. How would you implement this model efficiently?
>
> A. In equation (12), there are many weight multiplication. As the number of weight multiplication increases, slower the calculation speed is.
>
> In theoretical analysis, we analyzed using Gersgorin circle theorem similar to the paper "Recurrent Highway Network".

---

### Official Review · AnonReviewer3 · 2017-11-27
**Not exciting**

**Rating:** 4
**Confidence:** 4

**Review:**

The authors propose an RNN that combines temporal shortcut connections from [Soltani & Jang, 2016] and Gated Recurrent Attention [Chung, 2014]. However, their justification about the novelty and efficacy of the model is not well demonstrated in the paper. The experiment part is modest with only one small dataset Penn Tree Bank is used. The results are not significant enough and no comparisons with models in [Soltani & Jang, 2016] and [Chung, 2014] are provided in the paper to show the effectiveness of the proposed combination. To conclude, this paper is an incremental work with limited contributions.

Some writing issues:
1. Lack of support in arguments,
2. Lack of referencing to previous works. For example, the sentence “By selecting the same dropout mask for feedforward, recurrent connections, respectively, the dropout can apply to the RNN, which is called a variational dropout” mentions “variational dropout” with no citing. Or “NARX-RNN and HO-RNN increase the complexity by increasing recurrent depth. Gated feedback RNN has the fully connection between two consecutive timesteps” also mentions a lot of models without any references at all.
3. Some related papers are not cited, e.g., Hierarchical Multiscale Recurrent Neural Networks [Chung, 2016]

---

> ### Author Response · Authors · 2018-01-05
> **Response to AnonReviewer3**
>
>  We thanks the reviewers for their work. And your review was very helpful for me.
>
> Lack of support in arguments
>
> I added the reference papers belows.
> - Learning long-term dependencies in narx recurrent neural networks (NARX-RNN)
> - Higher order recurrent neural networks (HO-RNN)
> - Hierarchical multiscale recurrent neural networks
> - Memory augmented neural networks with wormhole connections (MANN)
> - A clockwork rnn

---

### Decision · Program_Chairs · 2018-01-29
**ICLR 2018 Conference Acceptance Decision**

**Decision:**

Reject

**Comment:**

meta score: 4
This paper concerns a variant to previous RNN architectures using temporal skip connections, with experimentation on the PTB language modelling task
The reviewers all recommend that the paper is not ready for publication and thus should be rejected from ICLR.  The novelty of the paper and its relation to the state-of-the-art is not clear.  The experimental validation is weak.
Pros:
 - possibly interesting idea
Cons:
 - weak experimental validation
 - weak connection to the state of the art
 - precise original contribution w.r.t state-of-the-art is not clear